# K-HATERS:
# A Hate Speech Detection Corpus in Korean with Target-Specific Ratings

**Chaewon Park** [†]    **Soohwan Kim** [‡]    **Kyubyong Park** [‡]    **Kunwoo Park** [† §]

[†]School of AI Convergence, Soongsil University
[‡]TUNiB
[§]Department of Intelligent Semiconductors, Soongsil University
kunwoo.park@ssu.ac.kr

## Abstract

Numerous datasets have been proposed to combat the spread of online hate. Despite these efforts, a majority of these resources are English-centric, primarily focusing on overt forms of hate. This research gap calls for developing high-quality corpora in diverse languages that also encapsulate more subtle hate expressions. This study introduces K-HATERS, a new corpus for hate speech detection in Korean, comprising approximately 192K news comments with target-specific offensiveness ratings. This resource is the largest offensive language corpus in Korean and is the first to offer target-specific ratings on a three-point Likert scale, enabling the detection of hate expressions in Korean across varying degrees of offensiveness. We conduct experiments showing the effectiveness of the proposed corpus, including a comparison with existing datasets. Additionally, to address potential noise and bias in human annotations, we explore a novel idea of adopting the Cognitive Reflection Test, which is widely used in social science for assessing an individual's cognitive ability, as a proxy of labeling quality. Findings indicate that annotations from individuals with the lowest test scores tend to yield detection models that make biased predictions toward specific target groups and are less accurate. This study contributes to the NLP research on hate speech detection and resource construction. The code and dataset can be accessed at https://github.com/ssu-humane/K-HATERS.

## 1 Introduction

Hate speech is defined by "offensive discourse targeting a group or an individual based on inherent characteristics (such as race, religion or gender)"[1]. This is a critical problem in society that can aggravate polarization and threaten democracy (Lorenz-Spreen et al., 2023). The massive volume of communication in the web and social media amplifies the negative impacts of hate speech. To curb its spread in online environments, previous NLP research has developed ML-based detection models. Labeled resources play a critical role in building supervised detection models; thus, a various form of offensive language and hate speech datasets have been proposed in the NLP community, and they facilitated the development of detection methods (Mathew et al., 2021; Kim et al., 2022b).

Despite the advancement in the online hate speech research, there are several rooms for improvement. **(1) Language**: Most of the existing datasets are limited to English and high-resourced languages (Davidson et al., 2017; Founta et al., 2018). It is critical to construct a non-English dataset for the development of detection models that are aware of culture-specific expressions. **(2) Implicitness**: The majority of previous research conducted on this subject has primarily concentrated on explicit manifestations of offensive language. A few studies investigated the implicit form of hateful content (ElSherief et al., 2021; Ocampo et al., 2023; Hartvigsen et al., 2022), most of which targeted English. **(3) Biased and noisy labels**: Manual annotation is a predominant approach for constructing labeled data. However, the labeling process could reflect inherent biases of human annotators, and it is hard to control the label quality (Sap et al., 2019; Sachdeva et al., 2022). Training a model on such a noisy and biased dataset can amplify biases by inference (Shah et al., 2020).

This paper aims to fill the three research gaps by presenting a new HATE speech detection corpus in Korean with target-specific RatingS, named **K-HATERS**. **(1)** Among Korean offensive language corpora, this resource represents the most extensive data collection to date, comprising 192,158 news comments and surpassing the size of the previously largest corpus by 1.75 times (Lee et al., 2022). **(2)** We design a rating scheme for target-specific and fine-grained offensiveness on a three-point Likert

---

[1]https://www.un.org/en/hate-speech/understanding-hate-speech/what-is-hate-speech

| Dataset | Size | Source | Labeling scheme | Target label | Hate rationale | Target rationale | Multi-scale ratings |
|---|---|---|---|---|---|---|---|
| BEEP! (Moon et al., 2020) | 9.3K | Naver news | A) none, offensive, hate
T) gender, others, none | | | | |
| KoLD (Jeong et al., 2022) | 40.4K | Youtube, Naver news | A) offensive, not
T) untargeted, individual, other, gender&sexual orientation, ethnicity&nationality, political affiliation, religion, miscellaneous | ✓ | ✓ | ✓ | |
| K-MHaS (Lee et al., 2022) | 109K | Naver news | A) hate speech, not hate speech
T) politics, origin, appearance, age, gender, religion, race, profanity | ✓ | | | |
| KODORI (Park et al., 2023) | 3.8K | Online communities | A) offensive, likely offensive, not offensive
T) not available | | | | ✓ |
| Ours | 192K | Naver news | A) level-2 hate, level-1 hate, offensive, normal
T) gender, age, race (origin), religion, politics, job, disability, individual, others | ✓ | ✓ | ✓ | ✓ |

Table 1: List of offensive language corpora in Korean (A: Abusive language categories, T: Target categories)

scale. It distinguishes between the explicit form of offensiveness (e.g., toxic expression) and the implicit form (e.g., sarcasm, stereotypes), facilitating a nuanced understanding of online hate. **(3)** We borrowed the idea of psychology and social science for improving labeling quality in the dataset construction. In particular, we propose using the cognitive reflection test (CRT) (Frederick, 2005), which was designed to assess a cognitive ability to suppress an intuitive wrong answer in favor of a reflective correct answer. Previous research in social science showed that the score is correlated to one's likelihood of following implicit biases and heuristics (Roozenbeek et al., 2022; Pennycook and Rand, 2021; Toplak et al., 2011). Our research hypothesis is that *the labels given by individuals with a low CRT score may be noisy and biased*. To test the hypothesis and support the effectiveness of the dataset, we conducted experiments with varying architectures and metrics that cover human-centered desiderata of hate speech classifiers, such as detection performance, fairness (Ramponi and Tonelli, 2022), and explainability (Mathew et al., 2021).

The key findings are summarized as follows:

1. We present a large-scale offensive language corpus of 192,158 samples along with target-specific offensiveness ratings on a 3-point Likert scale. The corpus provides offensiveness and target rationales as text spans.

2. We propose a label transformation method for the unified modeling of abusive language categories along with a multi-label target prediction. This method can be applied to existing corpora.

3. We investigate a novel approach of using the Cognitive Reflection Test to approximate labeling quality. Experimental results reveal that annotations from individuals with the lowest test scores can yield detection models that are both less accurate and unfairer than those derived from the control group.

4. A case study shows that hate speech is prevalent in the news comments posted on the politics, social, and world news section in a major news portal in South Korea. The hatred was mainly targeted at the groups of the attributes related to politics and regions.

## 2 Related works

Early studies used the binary label for classifying hate and normal text (Djuric et al., 2015; Badjatiya et al., 2017). Davidson et al. (2017) proposed using a taxonomy capturing a middle ground between hate speech and normal text. An abusive comment that is not hate speech is generally called *offensive*. A later study aimed to capture target-specific hate such as racism and sexism (Founta et al., 2018). Most research focused on overt forms of hate speech, but explicit hate is more easily identifiable, e.g., by lexicon-based methods (Davidson et al., 2017). Recent research focused on implicitness of hate speech and proposed new datasets (Jurgens et al., 2019; ElSherief et al., 2021; Wiegand et al., 2021; Ocampo et al., 2023; Nejadgholi et al., 2022; Hartvigsen et al., 2022). A study developed a taxonomy of implicit hate and provided a labeled dataset (ElSherief et al., 2021), but its annotation scheme is not generalizable in the Korean context (e.g., White Grievance).

In addition to prediction accuracy, recent studies focuses on human-centered desiderata of hate speech classifiers, such as explainability and fairness. Mathew et al. (2021) proposed a dataset that includes hate speech labels and annotators' rationale as a highlighted span. A recent study developed a pretraining method of masked rationale

prediction (Kim et al., 2022a). To promote the development of fair detection models, Röttger et al. (2021) introduced a functional test that measures a predictive bias of hate speech detection models. In the context of the Korean language, a handful number of resources is available, as shown in Table 1. BEEP! is the first dataset shared in the NLP community (Moon et al., 2020). The corpus consists of 9,381 news comments labeled either hate, offensive, or none, along with the bias label. Jeong et al. (2022) introduced KoLD by adopting a hierarchical labeling scheme. The dataset contains 40.4K online text along with labels on whether it is offensive and what is the target. Put it simply, KoLD employs a binary rating on whether a hate exists against a target group. Lee et al. (2022) proposed K-MHaS, a dataset of 109K comments of which the hate target is annotated by a multi-label scheme. As an alternative to such a crawling-based data collection, a recent study introduced a method that lets crowdworkers generate expressions (Yang et al., 2022). We introduce the largest and the first corpus along with target-specific offensiveness ratings, which contributes to the detection of non-explicit hate in Korean. While a recent corpus employed a three-class rating for offensiveness (Park et al., 2023), it lacks target labels and thus cannot be used to train a model that distinguishes between offensive expressions and hate speech.

## 3 Dataset

### 3.1 Data collection

We collected comments posted on news articles published via a major news portal in South Korea[2] over the period of July to August 2021. The target articles were collected from the society, world news, and politics sections, which are categorized as *hard news* (Lehman-Wilzig and Seletzky, 2010). They cover a newsworthy event to be of local, regional, national, or international significance, where active discussions and hate speech likely occur. We obtained 191,633 comments from 52,590 news articles, with an equal distribution of comments sourced from each section. To include a substantial amount of hateful comments in our labeled corpus, we trained a binary classifier on the BEEP! corpus (Moon et al., 2020) and sampled 95,816 comments from those with a sigmoid output of 0.5 or higher. 31,939 comments were randomly

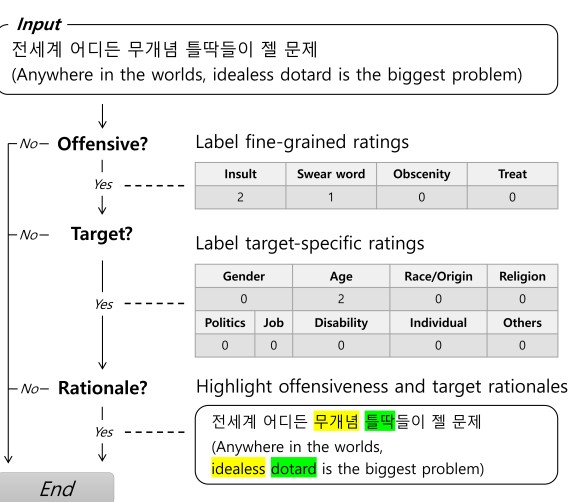

Figure 1: An illustration of labeling process

sampled from each section. Additionally, we included 8,367 comments from the BEEP! corpus that were published in the entertainments section, categorized as *soft news*. We included the dataset for the comparison of labeling schemes. In total, 200,000 news comments were prepared for label annotation.

### 3.2 Crowdsourced annotation

We used CashMission[3], a crowdsourcing service in Korea, to label the 200k news comments. Following a training session for 405 annotators hired by the company, we conducted pilot tests in which we ask the workers to label a small set of samples based on an initial guideline. The goal was to help the workers understand the task and for us to improve the guideline. We had a feedback session with each worker based on the annotation results. The final guideline includes several examples with a desired label and explanation derived from the pilot test.

**Labeling process** Figure 1 illustrates the labeling process. For each comment, an annotator was asked to label eleven offensiveness ratings. These ratings fall into two main categories: target-specific and fine-grained labels. The target-specific ratings measure the intensity of offensive expressions directed at one of the prominent hate targets in South Korea: gender, age, race (origin), religion, politics, job, and disability. The seven targets are categorized as protected attributes; thus, any offensive expressions against them are classified as hate speech (Zhang and Luo, 2019). To differentiate

---

[2]news.naver.com

[3]www.cashmission.com

| Value | Target-specific offensiveness ratings | | | | | | | | | Fine-grained ratings | | | |
| | GRP | | | | | | | IND | OTH | Insult | Swear words | Obscenity | Threat |
| | Gender | Age | Race | Religion | Politics | Job | Disability | | | | | | |
|---|---|---|---|---|---|---|---|---|---|---|---|---|---|
| 0 | 187,983 | 190,184 | 175,199 | 191,149 | 157,461 | 184,849 | 190,484 | 139,935 | 175,660 | 59,457 | 152,927 | 189,861 | 185,817 |
| 1 | 2,295 | 1,128 | 9,628 | 580 | 16,720 | 3,783 | 1,526 | 52,223 | 16,498 | 75,041 | 23,074 | 1,281 | 2,623 |
| 2 | 1,880 | 846 | 7,331 | 429 | 17,977 | 3,526 | 148 | - | - | 57,660 | 16,157 | 1,016 | 3,718 |

Table 2: Label distribution of thirteen rating variables

hate speech from offensive comments directed at individuals or unspecified groups, we introduced two binary variables. The first indicates if the comment is targeted at an individual, and the second denotes targeting unspecified groups. For simplicity, we refer to the seven targeted groups as GRP, individuals as IND, and unspecified targets as OTH. The fine-grained offensiveness ratings offer a more detailed insight, capturing specific sub-categories of offensiveness, such as insults, swear words, obscenities, and threats. We included these four ratings to facilitate a deeper, more nuanced understanding of offensive expressions.

Each rating is based on a 3-point Likert scale, ranging from 0 to 2. A value of 0 indicates no detected offensiveness. A value of 1 suggests mildly offensive or potentially offensive expressions, including sarcasm, stereotypes, or prejudice. A value of 2 denotes a clearly toxic expression likely to annoy readers. Besides assigning these scales, annotators were directed to highlight specific text spans that influenced their rating decision and the identified target of offensiveness. These highlighted spans are termed "offensiveness rationale" and "target rationale," respectively. The detailed guideline can be found in §A.2.

For the 200,000 comments, we employed rule-based filtering criteria[4] to identify and remove noisy labels. The final dataset comprises 192,158 comments, which was used for the experiments in the paper. Table 2 presents the label distribution. We observe the prevalence of samples with the value of 1. This finding suggests the importance of the proposed labeling scheme that distinguishes between normal and weakly offensive ratings. Without this multi-point rating approach, comments containing subtle forms of hate speech might be mistakenly labeled as normal without the multi-point rating scheme. We suspect that the content moderation on the platform — whether manual or automated — might have limited the occurrence of comments rated as 2.

Table 3 displays several examples along with

---

[4]detailed in §A.3

their English translations. These examples demonstrate that our rating scheme facilitates understanding both the intensity of offensive expressions and their respective targets. The last example highlights that a single text can contain multiple targets.

### 3.3 Cognitive reflection test

Biased annotators could inadvertently induce their implicit prejudices into the labels (Sap et al., 2019; Davidson et al., 2019; Sachdeva et al., 2022). If machine learning models are trained on such biased data, they potentially amplify these biases and generate harm by inference (Shah et al., 2020). To address the low-quality labeling issue and promote fairness in hate detection models, we suggest leveraging a psychological test as an indicator of labeling quality. Specifically, we propose the use of the Cognitive Reflection Test (CRT). The CRT was developed to assess an individual's cognitive capability to suppress an intuitive yet incorrect response in favor of a more reflective and accurate one (Frederick, 2005). This test is prevalently used across various disciplines in social science (Roozenbeek et al., 2022; Pennycook and Rand, 2021). Notably, studies have found correlations between CRT scores and implicit biases (Toplak et al., 2011). Accordingly, we set the following hypothesis:

> *H. The labels annotated by workers with a low CRT score, which implies that paid less attention to the labeling process, would be biased and noisy.*

To evaluate the hypothesis, we conducted the cognitive reflection test on all the annotators using a Korean variant of CRT comprising six questions (Kim, 2020). It details are provided in Table A1. Each participant's test score ranges from 0 to 6, where the participants get one point for every correct answer. We report the comparison results among different score groups in Section 6.

### 3.4 Label transformation

A straightforward solution for training a detection model using the proposed corpus would be to train

| Data example | Target-specific offensiveness ratings | | | | | | | | | Fine-grained ratings | | | |
| --- | --- | --- | --- | --- | --- | --- | --- | --- | --- | --- | --- | --- | --- |
| | GRP | | | | | | | IND | OTH | Insult | Swear words | Obscenity | Threat |
| | Gender | Age | Race | Religion | Politics | Job | Disability | | | | | | |
| 일본은 접종비용을 내나보네
I guess one should pay for getting vaccinated in Japan. | 0 | 0 | 0 | 0 | 0 | 0 | 0 | 0 | 0 | 0 | 0 | 0 | 0 |
| 금요일 월요일 생리휴가 쓰는 한녀 같누 ㅋㅋ
You look like a Korean girl, using her menstrual leave on Friday and Monday. | 2 | 0 | 0 | 0 | 0 | 0 | 0 | 0 | 0 | 2 | 2 | 0 | 0 |
| 전세계 어디든 무개념 틀딱들이 젤 문제
Anywhere in the world, idealess dotard is the biggest problem. | 0 | 2 | 0 | 0 | 0 | 0 | 0 | 0 | 0 | 2 | 0 | 0 | 0 |
| 순진한 호주인.,ㅋㅋ,한국인에.비하면 어리썩고 순진한 약간 모자란 느낌,,,
Innocent Australian haha, foolish and naive compared to Koreans, a little halfwitted. | 0 | 0 | 1 | 0 | 0 | 0 | 0 | 0 | 0 | 1 | 0 | 0 | 0 |
| 종교 에 미치 면 저리된다
That is what happens if one gets crazy about religion. | 0 | 0 | 0 | 2 | 0 | 0 | 0 | 0 | 0 | 1 | 2 | 0 | 0 |
| 기레기 야! 백신을 남을 위해서 맞나?
Hey such a presstitute! Are you getting a vaccine for someone else? | 0 | 0 | 0 | 0 | 0 | 2 | 0 | 0 | 0 | 1 | 0 | 0 | 0 |
| 페미니즘 정신병 남성연대 파이팅!
Feminism is mental illness, men's solidarity go go go! | 2 | 0 | 0 | 0 | 0 | 0 | 1 | 0 | 0 | 1 | 0 | 0 | 0 |

Table 3: Labeled data examples and their English translations (green highlight: offensiveness rationale, yellow highlight: target rationale, lime highlight: overlapped rationales)

a multi-label classification model that predict thirteen rating variables. Here, we propose an alternative modeling strategy based on an unified abusive language labeling scheme.

We transform the thirteen rating variables into two labels: one represents the four-class abusive language categories (ALC), and the other corresponds to multi-label target categories (TGT). The ALC label transformation process is illustrated in Figure 2. According to the community definition (Davidson et al., 2017), ALC broadly comprises three categories: *normal*, *offensive*, and *hate*.

- A comment is labeled as *normal* if all ratings of the comment are 0.

- A comment with an offensiveness rating above 0 directed at a target group is labeled as *hate*.

- If a comment has a rating above 0 but is not directed at a protected attribute group, it is classified as *offensive*.

Furthermore, we divide the hate labels into two tiers based on the highest rating value toward the target groups:

- A comment is labeled as *Level-2 hate* if its highest rating is 2 and it has labeled spans for the offensiveness rationale.

- A comment is deemed as *Level-1 hate* if the highest rating toward a protected attribute group is 1. Additionally, offensive comments without a specified rationale for offensiveness are categorized under level-1 hate. Given that

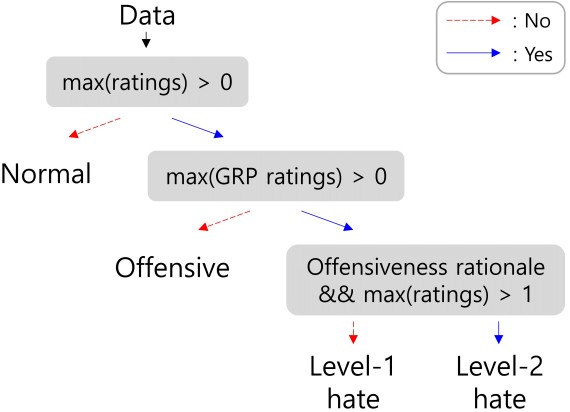

Figure 2: A flow chart for label transformation

| Data | Normal | Offensive | L1 hate | L2 hate |
| --- | --- | --- | --- | --- |
| Training | 46,643 | 70,467 | 19,537 | 35,511 |
| Validation | 2,709 | 4,093 | 1,135 | 2,063 |
| Test | 2,709 | 4,093 | 1,135 | 2,063 |
| Total | 52,061
(27.1%) | 78,653
(40.9%) | 21,807
(11.4%) | 39,637
(20.6%) |

Table 4: Label distribution of transformed abusive language categories across the split data

a rating of 1 indicates milder offensive expressions, such as sarcasm, stereotype, or prejudice, level-1 hate may encompass implicit hate expressions.

We split the dataset of 192,158 samples into 172,158/10,000/10,000 for training, validation, and test purposes, ensuring the transformed label distribution is maintained. Table 4 presents the distribution of ALC labels. Over 72% of the dataset contains offensive expressions, categorized as either offensive, level-1 hate, or level-2 hate. However, in many cases, annotators either could not identify

the target or the comments were not directed at a group of protected attributes. As a result, they were classified into offensive. Among the 61,444 hate expressions, a substantial portion of the comments, 21,807 comments (or 35.49%), were categorized as level-1 hate.

For TGT, we include a target if a corresponding rating is above 0. Thus, the maximum number of target categories a comment can have is nine. The TGT distribution can be referred to Table 2.

# 4 Detection experiments

## 4.1 Methods

We examine BERT fine-tuning methods for the detection of hate and offensive expressions using the transformed labeling scheme and raw rating variables, respectively.

**Using transformed labels**

- **H+T** is a *target*-aware method that employs target prediction as an auxiliary task. In addition to the classifier head for the 4-way abusive language category classification, the model includes a parallel dense layer that predicts nine logit scores, each of which corresponds to a multi-label target class (e.g., age, gender). The hidden layer is a 768-dimensional vector. Its training objective is to minimize $\mathcal{L}_{CE} + \alpha \mathcal{L}_{focal}$, where $\mathcal{L}_{CE}$ is the cross-entropy loss for the ALC prediction, $\mathcal{L}_{focal}$ is the focal loss (Lin et al., 2017) for multi-label target classification, and $\alpha$ is a hyperparameter.

- **H+T+R** additionally supervises the attention heads at the last layer by using the *rationale* spans. The training objective is to minimize $\mathcal{L}_{CE} + \alpha \mathcal{L}_{focal} + \beta \mathcal{L}_{Att}$, where $\mathcal{L}_{Att}$ is the mean of the cross-entropy loss measured for each token. $\alpha$ and $\beta$ are hyperparameters.

**Using rating variables**

- **H+T** predicts thirteen rating labels through a dense layer. The model is fine-tuned by optimizing the sum of cross-entropy for each label prediction, $\mathcal{L}_{CE:r}$. Given that the target variables include nine target-specific toxicity labels, we deem the model to be target-aware.

- **H+T+R** includes the attention supervision as an auxiliary objective. The training objective

is to minimize $\mathcal{L}_{CE:r} + \alpha \mathcal{L}_{Att}$, where alpha is a hyperparameter.

## 4.2 Evaluation metrics

We evaluate three desiderata of hate speech detection models.

**(1) Detection performance** We use micro and macro F1 for measuring how well a detection model performs. The greater value indicates the more accurate prediction. For the evaluation of models that predict thirteen rating variables, we transformed the predicted ratings using the process described in §3.3.

**(2) Fairness** We adopt the difference in the equalized odds ratio (Hardt et al., 2016) for understanding how equally a model is good at detection according to the hate target. We calculate a true positive rate and false positive rate respectively across different target groups and combine the target-wise nine values by the difference between the maximum and minimum values. The assumption is that a fair classifier should have a similar error rate across different target groups. We report the maximum value of class-wise odds as a unified measurement. The lower value indicates the more fair classifier.

**(3) Explainability** We evaluate the explainability of a classifier by using the two criteria introduced in earlier studies (Mathew et al., 2021): plausibility and faithfulness. A plausibility metric aims to quantify how plausible a given explanation is to humans (DeYoung et al., 2020). A faithfulness metric measures how precisely a given explanation reflects the reasoning process of a target model (Jacovi and Goldberg, 2020). Considering the normalized attention scores over the positions as a explainable mechanism of BERT classifier, we adopt the intersection-over-union (IOU) F1 for measuring plausibility. It measures the relative size of the matched tokens between the predicted and truth rationale spans. A span prediction is considered matched if the overlapped ratio with any of the truth span is larger than 0.5. The second explainability measurement is faithfulness, which aims to capture to what extent an explanation reflects the reasoning process of a model (Mathew et al., 2021). We instantiate faithfulness by the comprehensiveness metric that measures the decreased amount of prediction probability by removing predicted rationales from the text.

| Model | Detection performance | | Fairness | | Explainability | |
|---|---|---|---|---|---|---|
| | Micro F1 ↑ | Macro F1 ↑ | Equalized odds diff. (TPR) ↓ | Equalized odds diff. (FPR) ↓ | Plausibility ↑ | Faithfulness ↑ |
| **Using transformed labels** | | | | | | |
| **H+T** | **0.681**±**0.001** | **0.611**±**0.001** | **0.243**±**0.006** | **0.31**±**0.01** | 0.286±0.009 | 0.163±0.006 |
| **H+T+R** | 0.675±0.003 | 0.602±0.006 | **0.247**±**0.018** | 0.36±0.015 | 0.341±0.002 | **0.185**±**0.011** |
| **Using rating variables** | | | | | | |
| **H+T** | 0.663±0.001 | 0.6±0.002 | 0.316±0.006 | 0.338±0.017 | 0.239±0.019 | 0.094±0.012 |
| **H+T+R** | 0.663±0.001 | 0.6±0.001 | 0.330±0.003 | 0.338±0.013 | **0.351**±**0.002** | 0.123±0.007 |

Table 5: Performance of detection models. Mean and standard errors are reported.

## 4.3 Results

Table 5 presents the evaluation results of the four detection models, each differing in learning objectives and input data format. We make three observations. First, the H+T model with the transformed label achieved the best detection performance in terms of both micro and macro F1 scores. Second, the H+T model using transformed labels demonstrated enhanced fairness in its prediction results compared to that trained on rating variables. Among the models trained on the transformed labels, the H+T model obtained the lowest equalized odds difference score measured as FPR. This suggests that its predictions vary less across target groups compared to the other models. This observation contradicts the finding of a previous study (Mathew et al., 2021). The H+T and H+T+R models trained on the transformed labels showed similar performance when assessing the fairness metric via TPR. Third, there was no clear winner for explainability. While the H+T+R model trained on the transformed labels achieved the best results for the faithfulness metric. The H+T+R model trained on rating variables was the best for the plausibility metric. To summarize, our findings support the importance of employing transformed labels to enhance both the accuracy and fairness of hate speech detection. Consequently, we use the H+T model trained on the transformed label for the subsequent experiments.

## 5 Cross-dataset prediction

This section presents the results of a cross-dataset experiment to understand the generalizability of existing and proposed datasets, as summarized in Table 1. We trained a model on each dataset and then tested it on the test split of our dataset. To model an ALC category classifier, we applied a label transformation process, described in §3.4, to both KoLD and K-MHaS. Since the two datasets do not have a weakly offensive rating, the corresponding ALC label has three categories: normal, offensive, and hate. We sampled 8,367 instances, which is the size of BEEP!, from the training split of each dataset for a fair comparison.

Figure 3 presents the confusion matrix, with each axis representing the label categories. The models trained on BEEP!, KoLD, and K-MHaS achieved macro F1 scores of 0.528, 0.631, and 0.461, respectively. Notably, the classifier trained on BEEP! tends to misclassify offensive-labeled classes as hate labels. In contrast, the classifier trained on K-MHaS demonstrates an opposite trend: it frequently predicts normal labels for samples labeled as offensive and fails to predict hate labels for the level-1 hate samples. Meanwhile, the KoLD model exhibited predictive accuracy at 0.629, with its performance differing by only 0.04 compared to the intra-dataset model.

To gain deeper insights into the observed trends, we conducted a principal component analysis using a pretrained sentence transformer embedding[5]. Figure 4 depicts the distribution of the top two principal components for samples from each class across the four datasets. The results indicate that the labels from KoLD and our dataset are closely aligned. In contrast, K-MHaS and BEEP! exhibit distinct patterns. This divergence could account for the suboptimal performance observed in cross-dataset experiments.

## 6 Can CRT be a proxy of data quality?

This section examines the research hypothesis that labels given by workers with a low CRT score would be noisy and biased. To evaluate the hypothesis, we constructed two different training sets: (1) all samples annotated by the individuals with a

---

[5]https://huggingface.co/jhgan/ko-sbert-multitask

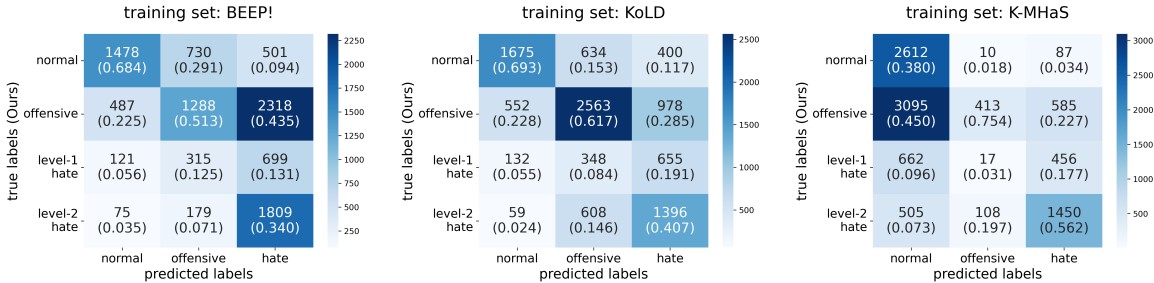

Figure 3: Cross-dataset prediction performance

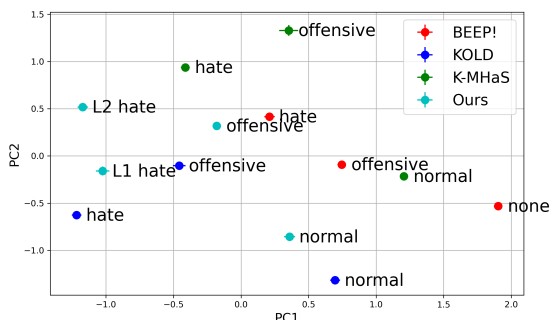

Figure 4: Principal component analysis of each label group across the datasets

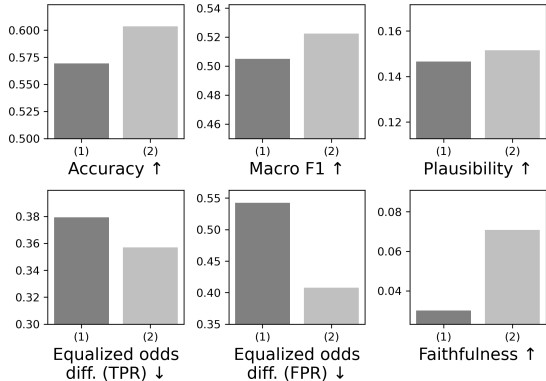

Figure 5: Performance difference between two models using different CRT score group annotations as training

score of 0 and (2) the rest samples. We randomly sampled the (2) dataset to the size of (1). By training an H+T model on each training set, we measured the six metrics of detection performance, fairness, and explainability. Figure 5 presents the evaluation results, suggesting that the model trained on the score-0 group is less accurate (in terms of accuracy and macro f1), more biased toward certain target groups (in terms of equalized odds differences), and less explainable (in terms of faithfulness). On the other hand, the performance margin is not significant for the plausibility. The finding supports the hypothesis in our dataset, implying the potential role of CRT for building the dataset with high-quality labels.

## 7  Case study: how many hates are prevalent in a Korean news portal?

We conduct a case study to understand the distribution of offensive language and hate speech in the wild. We applied the H+T model trained on the transformed unified labels to the unlabeled dataset of 191,663 news comments published in the society, world news, and politics sections shared over the period of July to August in 2021. The target dataset was randomly sampled from the initial collection, and it does not overlap with the proposed

corpus. For comparative analysis, we applied the classifier to the same number of comments from the unlabeled comments posted from the entertainment section, which was published concurrently with the BEEP! labeled corpus (Moon et al., 2020). This section falls under the *soft news* category(Lehman-Wilzig and Seletzky, 2010), in contrast to the hard news encompassed by our corpus. Results show that offensive and hate expressions are more prevalent in the comments associated with hard news than those from soft news. Among them, hate speech accounts for a larger fraction in the politics and worlds sections, targeted toward the politics and religion group as the major target (Figure 6b), respectively. By contrast, offensiveness embedded in comments from the entertainment section tend to be targeted toward individuals, such as celebrities.

## 8  Conclusion

This study introduces a new corpus for hate speech detection, representing the largest offensive language corpus in Korean to date. The dataset encompasses 192K news comments, each rated for target-specific offensiveness on a three-point Likert scale. For efficient modeling, we introduced a la-

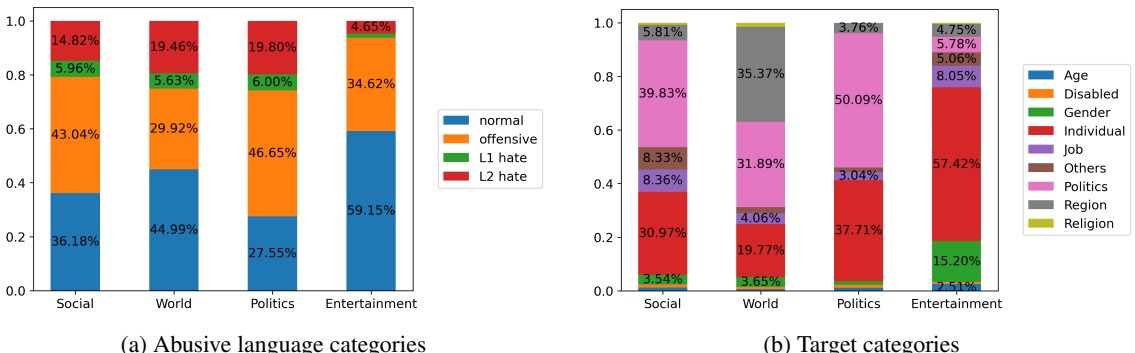

(a) Abusive language categories  (b) Target categories

Figure 6: Inference results on the unlabeled news comments

bel transformation strategy, aiming to unify labels for abusive language categories and multi-label target categories. Experimental results highlight the effectiveness of the proposed dataset and labeling scheme. Both the dataset and labeling strategy hold promise for advancing NLP research in hate speech detection.

Furthermore, inspired by studies in psychology and social science (Toplak et al., 2011; Roozenbeek et al., 2022; Teovanović et al., 2021; Pennycook and Rand, 2021), we tested the hypothesis that labels annotated by workers with low CRT scores—indicative of diminished attention during the labeling process—would be biased and noisy. Our findings revealed that using labels provided by annotators with the lowest test scores results in detection models that are both less accurate and unfairer. This discovery supports our hypothesis and suggests the CRT score's viability as a surrogate for annotation quality. To the best of our knowledge, this research is pioneering in integrating CRT into the process of ML dataset construction, offering a potential avenue of contribution to the wider NLP community.

## Limitations

First, the current research is centered on a single language, leaving the generalizability of the finding was not confirmed. It would be an exciting future direction to ascertain whether the findings, such as the effects of CRT, hold in other language and different cultural contexts. Second, while the study introduced a novel resource, it did not innovate on the detection method itself. Future studies could explore models that incorporate the results of the cognitive reflection test in their training. A straightforward application could be the integration of an instance-weighting method into the loss function.

Third, our resource was derived from the news comments in a single web portal. We targeted the platform because it is the major source of news consumption in South Korea, as frequently used in the construction of existing resources in Korea. Future studies could target hate expressions on emerging platforms, such as comments on short-form videos.

## Ethics and Impact Statement

This study introduces a novel offensive language corpus in Korean, comprising 192,158 samples. Among the publicly available offensive language corpora in Korean, our dataset stands out as the most extensive corpus and is the first one to use target-specific ratings on a 3-point Likert scale. This resource paves the way for subsequent research focused on the development of hate speech detection models in both Korean and multilingual settings. However, users should approach the dataset with caution, recognizing potential biases and risks. For instance, comments might exhibit biases towards certain political orientations based on user demographics. To promote responsible usage, we have included a data statement in the Appendix.

In this study, we explored the utility of CRT as a potential proxy for annotation quality. While our findings indicate a potential correlation between the CRT score and label quality, we caution against its use for filtering annotators during recruitment. Excluding individuals based on a low CRT score might inadvertently discriminate against specific groups. A potential use case of our finding might be to use the CRT score in pilot testing to gauge an annotator's attentiveness to the labeling process and discern any potential misunderstandings of the labeling guidelines. All participants in our study received equal compensation, irrespective of their CRT outcomes.

## Acknowledgments

The first two authors equally contributed to this work. This research was supported by the National Research Foundation of Korea (2021R1F1A1062691), the Institute of Information & Communications Technology Planning & Evaluation (IITP-2023-RS-2022-00156360), and DATUMO (SELECTSTAR) through the "2022 AI Dataset Supporting Business" program. Kunwoo Park is the corresponding author.

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

# A Appendix

## A.1 Data statement

We present the data statement for responsible usage (Bender and Friedman, 2018). The corpus

consists of 192,158 news comments consisting of 183,791 news comments collected by ourselves and 8,367 comments collected from a previous study (Moon et al., 2020). The collected dataset originates from the politics/world/social sections, which are categorized as hard news. The comments from the BEEP! corpus are from the entertainment section. The comment function in the entertainment section was disabled due to the prevalence of toxic expressions toward celebrities.

**Curation Rationale** We collected the raw data from news.naver.com, the largest news portal in Korea. We targeted news articles published in the society, world, and politics sections because discussions are active in the hard news.

**Language Variety** Our dataset consists of the news comments in Korean (ko-KR).

**Speaker Demographic** The user demographic is not available. However, considering that the portal site has the largest share of Korean, it can be assumed that speakers are mostly Korean.

**Annotator Demographic** The data annotation process was conducted on CashMission, a crowdsourcing platform in Korea. A total of 405 workers participated in an annotation. 21 workers are 10s, 222 workers are 20s, 116 workers are 30s, 35 workers are 40s, 9 workers are 50s, and 2 workers are 60s. In terms of gender, 309 women, 95 men and 1 none-binary participated.

**Speech Situation** News article in the hard news section deals with controversial events, so there are more likely to exist hate comments or toxicity comments. The target articles were published between July 2021 and August 2021. During that period, several highly contentious events unfolded, including the presidential election in South Korea, the Tokyo Olympics, the COVID-19 pandemic, and the Restoration of Taliban Control, to name a few.

**Text Characteristics** It includes hatred words restricted to Korea, such as hatred of specific political orientations and specific groups. For example, "대깨문" (a hate expression for the supporters of the former Korean president Moon), and "꼴페미" (a hate expression toward feminists).

## A.2 Detailed guideline

**Offensiveness scale (3-point ratings)** The degree of offensiveness is evaluated in a 3-point Likert scale: 0 (no offensiveness), 1 (weakly offensive, offensive-likely), 2 (obviously or seriously offensive). The value of 0 indicates there is no offensive expression. The value of 1 means there is room for disagreement about whether it is an offensive expression speech or not. This category includes implicit hate expressions, such as sarcasm, stereotypes of a target group, etc (ElSherief et al., 2021). The labeling decision could depend on the context of expression. This category of speech could not include any rationale. The value of 2 means that it is an explicitly offensive expression without any doubt. The comment can make readers very uncomfortable, implying that the comment should be censored in online platforms.

**Target-specific and fine-grained ratings (13 classes)** There are nine target-specific rating variables: Gender, Age, Race/Region, Disability, Religion, Politics (orientation), Job, Individual, Others. There are four fine-grained rating: Swear word/Curse, Insult, Treat/Violence/promoting crime, and Obscenity/Sexual harassment. All classes have more specific explanations and example comments. For example, in the case of 'Insult', it was explained as Demeaning and degrading the object's value and social evaluation, Expressing contemptuous feelings, and Disparaging others regardless of the facts. To give an example, "정말 참 한국 여자들 한심하다" (English Translation: "Korean women are really pathetic") can be a slight insult, and "쟤가 성병의 근원지였네" (English Translation: "He was the source of venereal disease") can be an obvious insult.

**Offensiveness rationale** Highlight the minimum span that will become a normal sentence if the corresponding span is masked or the span that will be included when building a hate expression dictionary. An expression of trying to avoid sanctions also should be highlighted. For example, explicit hate speech such as swear words, slang, and derogatory words, contextual hate speech, and inappropriate expressions such as sexual harassment should be highlighted.

**Target rationale** Target highlighting process highlights the target of offensive expressions. It can be overlapped with offensive representation highlighting. And this should also be highlighted by minimum span, the same as offensive representation highlighting. If the span containing the modifier can make the target of hate clearer, then the modifier should be included as the target span.

## A.3 Rule-based filtering

We excluded the data in the dataset if one of the following conditions is satisfied because the annotation is likely wrong.

1. Data with incorrect rationale annotation format.

2. Data where all fine-grained ratings have a value of 0, but IND or OTH ratings have a value of 1

3. Data with target-specific offensiveness ratings greater than 0 and a max rating greater than 1, but no rationale

4. Data with rationale beyond the max_length

## A.4 Detailed configuration

For experiments, we used KcBERT (Lee, 2020) as the BERT backbone based on comparisons with KPFBERT[6] and KLUE-BERT[7] (Table A2). The model is a BERT variant with 110M parameters, which was pretrained on a Korean corpus. We uses AdamW as an optimizer with a learning rate of 2e-5, and set the batch size as 32. Models were early-stopped using the validation set with the patience of 5. We repeated the model training five times with varying random seeds: 0,10,20,30,40. The computational environment consists of NVIDIA RTX A6000, 125G RAM, and AMD Ryzen Threadripper PRO 3975WX 32-Cores. Hyperparameters were optimized by random search on the validation set. $\alpha$ is 0.1 for attention supervision and 1 for training target labels. 2 attention heads were supervised for attention supervision.

## A.5 More analyses

**Inter-dataset similarity** To understand the representativeness of each corpus, we measured token-level and embedding similarities between the offensive language corpora in Korean. For the token-level similarity, we calculated the top-10000 tokens[8] in each corpus and measured the Jaccard similarity between each pair of the two token sets, as illustrated in Figure A1a. The results suggest that our dataset is the most similar to KoLD, and BEEP! is to K-MHas in terms of the token-level similarity. For the embedding similarity, we used a checkpoint

[6]https://huggingface.co/jinmang2/kpfbert
[7]https://huggingface.co/klue/bert-base
[8]by the BERT tokenizer

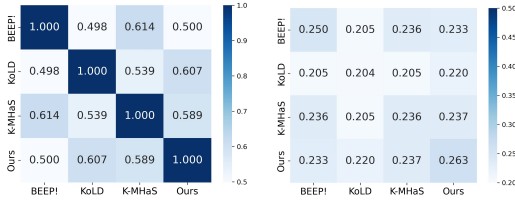

(a) Token-level similarity    (b) Embedding similarity

Figure A1: Similarity between offensive language corpora in Korean

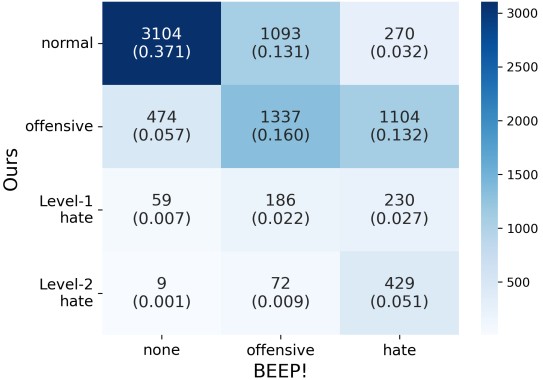

Figure A2: Label comparisons of the 8,367 samples in BEEP! and the proposed corpus

of BERT-based sentence transformer that is fine-tuned on the KorSTS and KorNLI dataset[9]. Figure A1b presents the results. The embedding analysis suggests that each corpus is not largely overlapped to other corpora in the embedding space.

**Label comparison** To evaluate the proposed labeling scheme, we included the labeled samples in the BEEP! dataset and examined how a sample is labeled in our dataset. Figure A2 presents the results. We observed that a large fraction of hate-labeled expressions in the BEEP! dataset were re-labeled as offensive in our dataset. We suspect that this is due to the lack of target labels in the previous corpus. Also, the level-1 hate samples in our corpus had been labeled in the BEEP! corpus as normal (59) or offensive (186). Altogether, the findings suggest the importance of target-specific ratings for the effective identification of hate.

**Cross-dataset prediction experiment** We conducted an experiment where we used a model trained on our own dataset to predict the labels of the previous corpus. For the BEEP!, similar to the label comparison, there is a tendency for hate data in the BEEP! dataset to be predicted as offen-

[9]https://huggingface.co/jhgan/ko-sroberta-multitask

| Question | Correct answer |
|---|---|
| 야구 배트와 야구공이 합쳐서 1달러 10센트다.
야구 배트가 야 구공보다 1달러 비싸다. 공은 얼마인가?
A bat and a ball cost 110 cents in total.
The bat costs 100 more than the ball. How much does the ball cost? | 5 cents |
| 의류 공장에서 5벌의 셔츠를 만드는데 5대의 기계를 사용해서 5 분이 걸린다.
100벌의 셔츠를 만들기 위해 100대의 기계를 사용 하면 몇 분이 걸릴까?
If it takes 5 machines 5 minutes to make 5 widgets,
how long would it take 100 machines to make 100 widgets? | 5 minutes |
| 연못에 있는 연꽃잎이 매일 2배 커진다. 연못 전체를 덮는데 48 일이 걸린다면,
연못의 절반을 덮는데 몇 일이 걸릴까?
In a lake, there is a patch of lily pads. Every day, the patch doubles in size. If it takes 48 days
for the patch to cover the entire lake, how long would it take for the patch to cover half of the lake? | 47 |
| 3명이 한 시간 당 3개의 장난감을 포장한다면, 2시간에 6개의 장난감을
포장하기 위해서는 몇 명이 필요한가?
If 3 people pack 3 toys per hour, how many people are
needed to pack 6 toys in 2 hours? | 3 |
| 영희의 중간고사 성적은 반에서 15번째로 높은 동시에 15번째 로 낮다.
영희의 반 학생은 모두 몇 명인가?
Younghee's midterm grades are the 15th highest and
15th lowest in her class. How many students are in Younghee's class? | 29 |
| 수영 팀에서 키가 큰 선수는 작은 선수에 비해 우승할 확률이 3배 높다.
올해 팀의 우승 횟수가 60번이라면, 키가 작은 선수는 60번 중 몇 번이나 우승했을까?
Tall swimmers on a swim team are three times more likely to win than shorter swimmers.
If the team won 60 championships this year, how many of those 60 did the short players win? | 15 |

Table A1: List of CRT questions and their English translations

| Model | Detection performance | | Fairness | | Explainability | |
|---|---|---|---|---|---|---|
| | Micro F1 ↑ | Macro F1 ↑ | Equalized odds diff. (TPR) ↓ | Equalized odds diff. (FPR) ↓ | Plausibility ↑ | Faithfulness ↑ |
| KcBERT | 0.589 | 0.55 | 0.267 | 0.271 | 0.285 | 0.205 |
| KPFBERT | 0.57 | 0.541 | 0.324 | 0.333 | 0.229 | 0.207 |
| KLUE-BERT | 0.573 | 0.543 | 0.324 | 0.359 | 0.129 | 0.173 |

Table A2: Effects of pretrained backbone models (Model: H+T with transformed labels)

sive. Additionally, in the test sets of KoLD and K-MHaS, a significant number of hate data were also predicted as offensive, which indicates that some of the hate data in the existing corpus target general groups or individuals rather than protected groups.

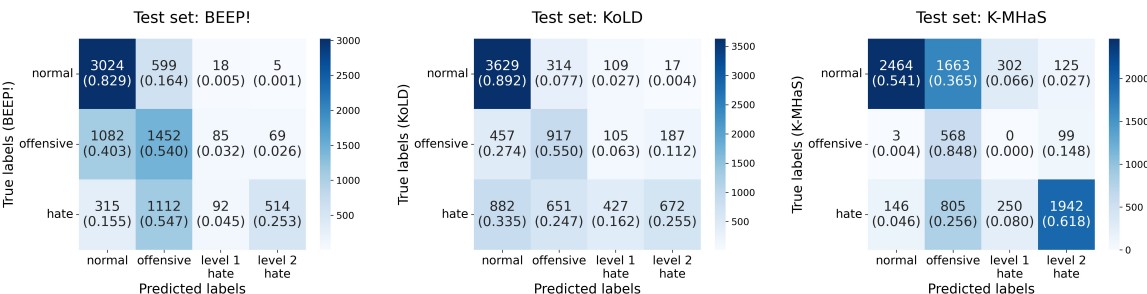

Figure A3: Cross-dataset prediction experiments that use our corpus for training