# OpenReview forum: "K-HATERS: A Hate Speech Detection Corpus in Korean with Target-Specific Ratings"
_EMNLP/2023/Conference — EMNLP 2023 Findings_

### Official Review · Reviewer_vzY9 · 2023-07-31

**Soundness:** 3

**Excitement:**

2: Mediocre: This paper makes marginal contributions (vs non-contemporaneous work), so I would rather not see it in the conference.

**Missing References:**

The article lacks references to important works that would allow relevant comparisons to support the scientific quality of the work done by the authors.

For what concerns a general scenario there are surveys that should be cited also to better introduce the meaning of the notion of hate speech and other ones closely related (such as offensiveness, aggressiveness, stereotype ...), for example, Poletto et al. (2020) Resources and benchmark corpora for hate speech detection: a systematic review. Language resources and evaluation 55 or Fortuna et al. (2018) A Survey on Automatic Detection of Hate Speech in Text. ACM computing survey 51: 4.

For what concerns the datasets and results, it should be important to cite at least some of the experiences done in the context of evaluation campaigns. This is the context where the datasets became benchmarks for the community and where the state-of-the-art is established for tasks such as hate speech detection. Moreover, the papers published in this kind o context are the most cited by the community, see for instance, Basile et al. (2019) SemEval-2019 Task 5: Multilingual Detection of Hate Speech Against Immigrants and Women in Twitter. In Proceedings of the 13th International Workshop on Semantic Evaluation or other similar foundational works.
.

**Paper Topic And Main Contributions:**

The paper introduces a novel large corpus for Korean in which hate speech is annotated using a fine-grained scheme.
The corpus consists of news comments and the scheme includes the rating of offensiveness, the identification of the target and the rationale, while the annotation has been performed by 405 annotators of a crowdsourcing platform.
Finally, the paper describes the results achieved by using different models trained on this corpus.

**Questions For The Authors:**

In the introduction, implicitness is cited as a topic addressed in the introduction of the paper.
Nevertheless, it is not clear how the implicit forms of hate speech are distinguished from the explicit ones. Are the authors assuming that all their data are implicit forms of hate speech? This should be motivated based on some criterion or analysis.
In the annotation, implicitness is not annotated, while in the experiments, the results are not organized to show how the models perform on implicit forms of hate rather than on explicit ones. This aspect is in effect interesting and should be better dealt with in the paper.

As far as the annotation process: How many annotations are collected for each message? it has been evaluated the disagreement among the annotators? it is quite unusual that the problem of bias is mentioned when a comparison between annotators is not made that allows an effective evaluation, even if only on a sample of data.

Another point that should be made more clear is the annotation bias. The CRT score is a measure of the attention paid by the annotator to the annotation task, but it is clear if there is a relationship between it and the bias. At several points (e.g. at the end of section 6) it is cited the bias of the annotators but how it is evaluated? Also considering that the annotation process does not involve evaluation of the disagreement.




**Reasons To Accept:**

The most important contribution of this paper is a novel resource for Korean.
In principle, several interesting aspects are considered (but not discussed enough) among which there are the implicitness of hate speech or the problem of bias in the annotation.
Also, the annotation scheme can be of interest. Its novelty consists of the categories to be annotated, in the organization in a cascade of them and the consideration of the CRT score in the evaluation of annotators.

**Reasons To Reject:**

The discussion about implicitness is not convincing for the following reasons:
1)  it cannot be truly said that the majority of previous works concentrated on explicit forms of hate speech because the majority of works about hate speech detection are based on corpora collected from social media, that is environments where a large portion of explicit hate is automatically detected or signalled by users to be immediately removed (and therefore often not available in the data collected by researchers);
2) the authors say that implicitness consists of irony or sarcasm; also not considering that I disagree with this assumption (according to several other researchers), a discussion about this point is not given in the paper; there are several works in which irony and sarcasm are discussed and resources in which those phenomena are annotated and a comparison with them is missing;
3) also in the annotation and in the evaluation implicitness is not considered, nor some distinction is proposed between implicit and explicit forms of hate.

The evaluation section does not provide a comparison with the results achieved by other existing models or by some baseline models. The literature cited by the authors does not include important works such as those obtained in evaluation campaigns, for example, in SemEval tasks.

It is not clear from the paper whether the dataset will be made available to the research community.

**Reproducibility:**

3: Could reproduce the results with some difficulty. The settings of parameters are underspecified or subjectively determined; the training/evaluation data are not widely available.

**Reviewer Confidence:**

3: Pretty sure, but there's a chance I missed something. Although I have a good feel for this area in general, I did not carefully check the paper's details, e.g., the math, experimental design, or novelty.

**Typos Grammar Style And Presentation Improvements:**

I did not find typos or grammatical errors. In my opinion (as a non-native English speaker), the paper seems well-written and organized.

---

> ### Author Rebuttal · Authors · 2023-08-29
>
> We thank the reviewer for the time and effort for this constructive review.
> - “Reasons To Reject” #1: While automatic and manual moderation can reduce the volume of explicit hate expressions in online platforms, since moderation tools are imperfect, there can be many toxic expressions left available, as empirically observed in our dataset (Table 4). Level-2 Hate represents the explicit hate expressions.
> - “Reasons To Reject” #2: Irony (or sarcasm) is the main subclass of implicit hate identified in previous research (e.g., ElSherief et al., 2021). In the dataset construction, we identified many examples of implicit hate that is expressed through sarcasm, stereotype, or prejudice.
> - “Reasons To Reject” #3: In the annotation, we asked the annotator to label 1 if the target comment includes weakly toxic expressions such as sarcasm, stereotypes, or prejudices. They were instructed to label higher scales if a target comment includes explicit toxic expressions. Following the transformation process (Figure 2), we deem a comment with a rating of 1 toward a group rating as an implicit hate expression. We will clarify the difference in Section 3.4.
> - The dataset and code will be available through a GitHub repository if this paper is accepted.
> - “Are the authors assuming that all their data are implicit forms of hate speech?”: No. We designed the annotation scheme and transformation process to make Level-2 Hate represent the explicit form of hate expressions.
> - “How many annotations are collected for each message?”: A single annotator was assigned to each comment. To ensure the quality, we conducted pilot tests. Each round uses 100 examples. After each round of pilot test annotation, they reported unclear cases to the authors and were instructed. They retook a test with a different set of examples. The process was repeated until the annotator passed the test and found no ambiguities.  They were paid for the pilot tests with the same conditions of the main annotation task. The results of the cross-dataset experiments (Section 5) suggest the high quality of our dataset.
> - How annotator bias was measured: We did not measure the annotator bias explicitly. It was measured by a proxy measure in classification experiments, i.e., the fairness evaluation of classifiers. Our assumption is that if a model is trained on biased annotations, the model would lead to worse fairness.
> - Missing references: Thank you for the suggestion. We will add the papers to the updated version.

---

### Official Review · Reviewer_dMtf · 2023-08-04

**Soundness:** 3

**Excitement:**

3: Ambivalent: It has merits (e.g., it reports state-of-the-art results, the idea is nice), but there are key weaknesses (e.g., it describes incremental work), and it can significantly benefit from another round of revision. However, I won't object to accepting it if my co-reviewers champion it.

**Paper Topic And Main Contributions:**

This paper introduces a novel manually-annotated Target-specific Hate Speech Detection dataset sourced from a prominent news portal in South Korea. In comparison to existing datasets, this proposed dataset serves as a relevant testbed for evaluating the effectiveness of Target-specific Hate Speech Detection methods in real-world scenarios. The authors conduct a comprehensive exploration of the dataset across various scenarios and present intriguing avenues for future research. Through their experiments, they demonstrate the formulation of a multitask problem as a hate speech detection task, leveraging large pre-trained sequence-to-sequence models. Notably, the authors conduct a case study that reveals hate speech's higher prevalence in news comments posted on hard news than soft news.

**Questions For The Authors:**

- What is "the current largest corpus" in line 58?
- "We present a large-scale offensive language corpus of 192,158 samples..." I think this is one of the key contribution of the paper. It is good idea to provide the link of the available data and codes in the manuscript.
- p3, line 170, 'including the same size of data '. what does the same size mean?
- Please elaborate on the annotators' qualifications, the payment for the labelling task, and a clearer description of "a small set of samples" in Section 3.2.
- While the authors have provided a comprehensive overview of the target-specific hate ratings task, additional information on the linguistic aspects of the task would be beneficial. This would help readers to understand the theoretical foundations of the study and the methods used to achieve the results.
- The authors have presented a benchmark dataset for target-specific hate speech detection, but there is no explanation of how they assessed the dataset's quality or determined the Inter Annotator Agreement (IAA).
- The texts and numbers in Figure 6 are not clearly visible. Please improve the quality of this figure. Additionally, the authors should discuss the specific reasons behind misclassifications of a certain class compared to others.
- To aid researchers in understanding the benefits and limitations of the Baseline Models for target-specific hate ratings used in this study, it would be valuable for the authors to explain their decision-making process behind their selection.
- The proposed Label transformation solution includes Normal, Offensive, Level-1 hate, and Level-2 hate labels. However, the difference between Level-1 and Level-2 is not clearly explained. Further clarification is needed regarding the character, impact, and significance of this label transformation.



**Reasons To Accept:**

- The paper's organization and presentation are commendable overall.
- The work exhibits strong performance, with promising results in the hate speech detection task.
- The paper introduces specific benchmark datasets tailored to a particular task.

**Reasons To Reject:**

- While the authors have provided a comprehensive overview of the target-specific hate ratings task, additional information on the linguistic aspects of the task would be beneficial. This would help readers to understand the theoretical foundations of the study and the methods used to achieve the results.
- Please elaborate on the annotators' qualifications, the payment for the labelling task, and a clearer description of "a small set of samples" in Section 3.2.
- The authors have presented a benchmark dataset for target-specific hate speech detection, but there is no explanation of how they assessed the dataset's quality or determined the Inter Annotator Agreement (IAA).
- The authors state that their dataset is the largest among current Korean datasets, with over 200k+ samples. However, no experiments or analyses have been conducted to support this claim. It is essential to clarify whether such a large dataset is necessary for effectively addressing this problem.
- The texts and numbers in Figure 6 are not clearly visible. Please improve the quality of this figure. Additionally, the authors should discuss the specific reasons behind misclassifications of a certain class compared to others.
- To aid researchers in understanding the benefits and limitations of the Baseline Models for target-specific hate ratings used in this study, it would be valuable for the authors to explain their decision-making process behind their selection.
- The proposed Label transformation solution includes Normal, Offensive, Level-1 hate, and Level-2 hate labels. However, the difference between Level-1 and Level-2 is not clearly explained. Further clarification is needed regarding the character, impact, and significance of this label transformation.


**Reproducibility:**

3: Could reproduce the results with some difficulty. The settings of parameters are underspecified or subjectively determined; the training/evaluation data are not widely available.

**Reviewer Confidence:**

5: Positive that my evaluation is correct. I read the paper very carefully and I am very familiar with related work.

**Typos Grammar Style And Presentation Improvements:**

- The authors should explain the meaning of the abbreviations in Figure, Table such as IND, OTH to ensure that readers understand the information presented. Abbreviations need to be explained right in the caption, instead of only discretely mentioned in the text.
- As data preprocessing is a crucial step in natural language processing tasks, the authors should include details on the data preprocessing techniques used to ensure the reproducibility of the study.

---

> ### Author Rebuttal · Authors · 2023-08-29
>
> We thank the reviewer for the time and effort for this constructive review.
> - Linguistic aspects: Due to the lack of space, we did not include the analysis, but it would be an exciting direction to understand the linguistic patterns of each category.
> - Annotator payment: Following the agency’s guidelines, annotators were paid $0.05 per example. Each annotator was assigned ~100 examples that were sampled from 10,000 comments. Each comment was assigned to three to four annotators.
> - IAA: We did not measure IAA for the entire set because each example was assigned to a single annotator. To ensure the quality, we conducted pilot tests. Each round uses 100 examples. After each round of pilot test annotation, they reported unclear cases to the authors and were instructed. They retook a test with a different set of examples. The process was repeated until the annotator passed the test and found no ambiguities. They were paid for the pilot tests with the same conditions of the main annotation task. The results of the cross-dataset experiments (Section 5) suggest the high quality of our dataset.
> - The currently available largest corpus is K-MHaS, comprising 109.6K comments. Table 1 presents the size of existing corpora. The advantage of using a large dataset is two-fold. First, the larger resource would cover more diverse patterns of each class, and therefore, a model trained on the corpus would be more representative. Second, a larger corpus allows for training larger models with more parameters. The model prediction would be more accurate.
> - Figure 6: We will increase the visibility of the embedded text/numbers.
> - Model selection: We set the standard classifier model that tackles the target task (H+T). We added the rationale training as an auxiliary objective for H+T+R as proposed in previous studies (e.g., Mathew et al., 2021)
> - Clarification on label transformation: Level-2 Hate represents explicit toxic expressions toward one or multiple groups, which has a value of 2 for the corresponding group ratings. Level-1 Hate represents weakly toxic expressions toward one of the multiple target groups, which has a value of 1 for one of the target group ratings and the maximum value is also 1. Therefore, Level-2 Hate consists of explicit hate expressions, and Level-1 Hate comprises implicit hate expressions. The label transformation process allows for detecting implicit expressions more effectively, as presented in Table 5. Section 5 presents the effectiveness of the proposed transformation process. Section 7 shows the prevalence of implicit hate expressions in online comments, which may cause online harm and undermine the platform's longevity. The proposed dataset and label transformation enable the detection of both implicit and explicit forms of hate. We will update Section 3.4 in the next version to help potential readers understand the difference more clearly.
> - We submitted the sample dataset and code to the openreview submission. A URL for the GitHub repository will be included in the final version if this paper is accepted.
> - Thank you for the comments on the “Typos Grammar Style And Presentation Improvement.” We will update the paper according to your suggestion.

---

### Official Review · Reviewer_9cnA · 2023-08-04

**Soundness:** 3

**Excitement:**

4: Strong: This paper deepens the understanding of some phenomenon or lowers the barriers to an existing research direction.

**Paper Topic And Main Contributions:**

The paper presents a new dataset of hate/offensive speech in Korean, with the novelty of labeling different scales of offensiveness in order to catch the implicitness of the offensive language.

The novelty of the paper is grounded in the work on Korean and the labeling schema, which is complemented with the cognitive reflection test for filtering out skewed annotators. However, I see that the main point of the paper is the new dataset and not the experiments. For this reason, I have some concerns:

1. I'm confused with the method to get the comments. In section 3.1, it is said that there are some comments from the dataset BEEP!. Why did you take comments from other dataset? I do not really understand this point.

2. Did you study the inter-agreement among annotators? If you used so much annotators, it maybe impractical the calculation of the inter-annotator agreement, but beyond the used of the CRT test, did you use an additional assessment of the quality of the annotation.

I think that this two points are more important than the classification results, and I miss a larger description of the details of the new dataset.

**Questions For The Authors:**

- Why did you take comments from other dataset?
- Did you study the inter-agreement among annotators?

**Reasons To Accept:**

- The novelty of a new dataset for hate speech classification in Korean.
- The use of the cognitive reflection test to identify skewed annotators.

**Reasons To Reject:**

- The lack of details of the building process of the dataset.
- The lack of evaluation of the agreement among annotators.

**Reproducibility:**

2: Would be hard pressed to reproduce the results. The contribution depends on data that are simply not available outside the author's institution or consortium; not enough details are provided.

**Reviewer Confidence:**

4: Quite sure. I tried to check the important points carefully. It's unlikely, though conceivable, that I missed something that should affect my ratings.

---

> ### Author Rebuttal · Authors · 2023-08-29
>
> We thank the reviewer for the time and effort for this constructive review.
> - We have two reasons to include the BEEP! dataset in our corpus.
>   1. BEEP! comprises the comments on the Entertainment section. Due to the cyberbullying issues toward celebrities, the comment features have been disabled for the section since 2020. The inclusion of the comments increases the topical diversity of our corpus. Our data collection covers news comments on hard news: society, world news, and politics sections.
>   2. The labels on the BEEP! dataset were used for verifying the quality of our corpus. The label comparison results are in Figure A3.
> - We did not measure IAA for the entire set because each example was assigned to a single annotator. To ensure the quality, we conducted pilot tests. Each round uses 100 examples. After each round of pilot test annotation, they reported unclear cases to the authors and were instructed. They retook a test with a different set of examples. The process was repeated until the annotator passed the test and found no ambiguities. They were paid for the pilot tests with the same conditions of the main annotation task. The results of the cross-dataset experiments (Section 5) suggest the high quality of our dataset.
> - We believe the dataset and paper can make a broad contribution to those working on hate speech and resource construction by the largest corpus in Korean and new findings on CRT. Thank you for your constructive feedback.

---

### Meta-Review · Area_Chair_HfQj · 2023-09-15

**Recommendation:** 4

**Metareview:**

The paper introduces a unique dataset focused on hate and offensive speech in Korean. This dataset is distinct due to its detailed labeling system that measures varying degrees of offensiveness, aiming to capture both explicit and implicit offensive language. The data is sourced from a major South Korean news portal, and its annotations are designed to identify the target of the hate speech, the rationale behind it, and the level of offensiveness. To ensure the quality of annotations, the dataset employs a cognitive reflection test to filter out potentially biased annotators. The dataset comprises news comments, and a significant number of annotators (405) from a crowdsourcing platform have been involved in the annotation process. The paper also explores the application of this dataset in hate speech detection tasks, using advanced models. A case study within the research indicates a higher occurrence of hate speech in hard news comments compared to soft news. However, there are concerns about the dataset's sourcing and the inter-annotator agreement process.

The paper presents a novel dataset for hate speech classification in Korean, filling a gap in the research landscape. The dataset's uniqueness is further emphasized by its detailed annotation scheme, which captures the implicit nuances of hate speech. This scheme is not only innovative in the categories it annotates but also in its hierarchical organization. Another significant contribution is the use of the cognitive reflection test (CRT) to filter out biased or skewed annotators, ensuring the quality and reliability of the dataset. The paper is well-organized, with clear presentation and structure, making it accessible to readers. The research demonstrates strong performance in the hate speech detection task, providing promising results that can serve as benchmarks for future studies. Overall, the paper's contributions, particularly the introduction of a new resource for Korean and the thoughtful consideration of potential biases in annotation, make it a valuable addition to the field.

One of the main issues is that the authors' discussion on implicit hate speech is unconvincing. They overlook the fact that many hate speech studies are based on social media data, where explicit hate is often removed. Their definition of implicitness, equating it to irony or sarcasm, is not universally accepted and lacks a thorough discussion. Moreover, the paper doesn't differentiate between implicit and explicit forms of hate in its annotations or evaluations.

The authors and the reviewers interacted during the rebuttal period to fix some other issues indicated by the reviewers.

---

### Decision · Program_Chairs · 2023-10-07

**Decision:**

Accept-Findings

**Comment:**

The paper introduces a unique dataset focused on hate and offensive speech in Korean. This dataset is distinct due to its detailed labeling system that measures varying degrees of offensiveness, aiming to capture both explicit and implicit offensive language. The data is sourced from a major South Korean news portal, and its annotations are designed to identify the target of the hate speech, the rationale behind it, and the level of offensiveness. To ensure the quality of annotations, the dataset employs a cognitive reflection test to filter out potentially biased annotators. The dataset comprises news comments, and a significant number of annotators (405) from a crowdsourcing platform have been involved in the annotation process. The paper also explores the application of this dataset in hate speech detection tasks, using advanced models. A case study within the research indicates a higher occurrence of hate speech in hard news comments compared to soft news. However, there are concerns about the dataset's sourcing and the inter-annotator agreement process.

The paper presents a novel dataset for hate speech classification in Korean, filling a gap in the research landscape. The dataset's uniqueness is further emphasized by its detailed annotation scheme, which captures the implicit nuances of hate speech. This scheme is not only innovative in the categories it annotates but also in its hierarchical organization. Another significant contribution is the use of the cognitive reflection test (CRT) to filter out biased or skewed annotators, ensuring the quality and reliability of the dataset. The paper is well-organized, with clear presentation and structure, making it accessible to readers. The research demonstrates strong performance in the hate speech detection task, providing promising results that can serve as benchmarks for future studies. Overall, the paper's contributions, particularly the introduction of a new resource for Korean and the thoughtful consideration of potential biases in annotation, make it a valuable addition to the field.

One of the main issues is that the authors' discussion on implicit hate speech is unconvincing. They overlook the fact that many hate speech studies are based on social media data, where explicit hate is often removed. Their definition of implicitness, equating it to irony or sarcasm, is not universally accepted and lacks a thorough discussion. Moreover, the paper doesn't differentiate between implicit and explicit forms of hate in its annotations or evaluations.

The authors and the reviewers interacted during the rebuttal period to fix some other issues indicated by the reviewers.